# Clinical Evaluation of Direct Reverse Transcription PCR for Detection of SARS-CoV-2 Compared to Conventional RT-PCR in Patients with Positive Rapid Antigen Test Results during Circulation of Emerging Viral Variants

**DOI:** 10.3390/diagnostics13243668

**Published:** 2023-12-14

**Authors:** Ming-Jr Jian, Chi-Sheng Chen, Hsing-Yi Chung, Chih-Kai Chang, Cherng-Lih Perng, Hung-Sheng Shang

**Affiliations:** 1Division of Clinical Pathology, Department of Pathology, Tri-Service General Hospital, National Defense Medical Center, Taipei 114, Taiwan; mj0106@gmail.com (M.-J.J.); tddcsc@mail.ndmctsgh.edu.tw (C.-S.C.); cindyft12@gmail.com (H.-Y.C.); mblkaiser@gmail.com (C.-K.C.); ponchenli@gmail.com (C.-L.P.); 2Graduate Institute of Medical Science, National Defense Medical Center, Taipei 114, Taiwan

**Keywords:** COVID-19, antigen test, direct RT-PCR, omicron variant, at-home kit

## Abstract

The emergence of the Omicron (B.1.1.529) variant of SARS-CoV-2 has precipitated a new global wave of the COVID-19 pandemic. The rapid identification of SARS-CoV-2 infection is imperative for the effective mitigation of transmission. Diagnostic modalities such as rapid antigen testing and real-time reverse transcription polymerase chain reaction (RT-PCR) offer expedient turnaround times of 10–15 min and straightforward implementation. This preliminary study assessed the correlation between outcomes of commercially available rapid antigen tests for home use and conventional reverse transcription polymerase chain reaction (RT-PCR) assays using a limited set of clinical specimens. Patients aged 5–99 years presenting to the emergency department for SARS-CoV-2 testing were eligible for enrollment (*n* = 5652). Direct PCR and conventional RT-PCR were utilized for the detection of SARS-CoV-2. The entire cohort of 5652 clinical specimens was assessed by both modalities to determine the clinical utility of the direct RT-PCR assay. Timely confirmation of SARS-CoV-2 infection may attenuate viral propagation and guide therapeutic interventions. Additionally, direct RT-PCR as a secondary confirmatory test for at-home rapid antigen test results demonstrated sensitivity comparable to conventional RT-PCR, indicating utility for implementation in laboratories globally, especially in resource-limited settings with constraints on reagents, equipment, and skilled personnel. In summary, direct RT-PCR enables the detection of SARS-CoV-2 with a sensitivity approaching that of conventional RT-PCR while offering expedient throughput and shorter turnaround times. Moreover, direct RT-PCR provides an open-source option for diagnostic laboratories worldwide, particularly in low- and middle-income countries.

## 1. Introduction

Cases of pneumonia caused by the novel severe acute respiratory syndrome coronavirus-2 (SARS-CoV-2) were reported in 2020. Subsequently, the COVID-19 pandemic ensued and is still ongoing due to the emergence of several variants of concern (VOCs). The initial outbreak began in Wuhan, China, in December 2019 and rapidly spread globally, with the World Health Organization declaring COVID-19 a pandemic on 11 March 2020. Multiple SARS-CoV-2 variants have emerged over the course of the pandemic, including Alpha, Beta, Gamma, Delta, and Omicron, which have demonstrated increased transmissibility and/or immune evasion compared to earlier strains. Persistent transmission worldwide has been fueled by factors such as vaccine hesitancy and inequitable vaccine access globally. As of November 2022, over 6.6 million COVID-19 deaths had been reported worldwide, underscoring the sustained impact of the pandemic. Continued genomic surveillance, the development of updated vaccines, and equitable vaccine distribution remain crucial to counter the ongoing threat posed by emerging VOCs [1,2,3,4,5]. To December 2022, more than 650 million cases of COVID-19 had been reported, with a global mortality of over 6 million (https://covid19.who.int/). RNA polymerases lack proofreading activity; hence, RNA viruses such as SARS-CoV-2 typically have higher mutation rates than DNA viruses, resulting in the evolution of new genetic variants. The SARS-CoV-2 virus has continued to evolve during the COVID-19 pandemic, with the emergence of variants that demonstrate increased transmissibility, immune evasion, and virulence compared to ancestral strains. The first variants of concern, Alpha and Beta, emerged in late 2020 and featured multiple mutations of the spike protein. In 2021, the Delta variant became globally dominant, followed by Omicron lineages like BA.1 and BA.5, which have extensive mutations in the receptor binding domain of the spike protein. Viral evolution is enabled by SARS-CoV-2′s error-prone RNA polymerase, which causes approximately 1–2 mutations per month due to lack of proofreading capacity. The combination of a high replication rate and ongoing transmission in large, naïve populations facilitates the generation of mutations and selection of fitter variants. Continued genomic surveillance and the development of updated vaccines tailored to emerging variants remain key to overcoming the public health threat posed by SARS-CoV-2 evolution [6,7]. The newly arising mutations are rapidly increasing and becoming dominant, conferring a competitive advantage with respect to viral replication, transmission, or immune evasion [8,9]. Omicron B.1.1.529 (later named BA.1) was first detected in South Africa in November 2021 and rapidly spread worldwide [10,11,12,13]. Since its emergence, Omicron has evolved into several major sub-lineages, with BA.1 as the initial dominant Omicron sub-lineage. By early 2022, the BA.2 sub-lineage began increasing in prevalence globally [14]. Later in 2022, additional Omicron sub-variants like BA.4 and BA.5, containing spike mutations beyond BA.1/BA.2, became globally dominant. The rapid succession of Omicron sub-lineages, each with mutations augmenting transmissibility, immune evasion, or virulence compared to its predecessors, underscores the ongoing threat of SARS-CoV-2 evolution [15,16].

Throughout the COVID-19 pandemic, nucleic acid amplification tests (NAATs) and rapid antigen tests (RATs) have been primarily used to diagnose ongoing SARS-CoV-2 infection. NAATs, such as reverse transcription polymerase chain reaction (RT-PCR), directly detect viral RNA and are considered the gold standard for diagnostic sensitivity and specificity. However, NAATs require specialized laboratory equipment and trained personnel, resulting in longer turnaround times of 1–3 days for results. Alternatively, RATs detect the SARS-CoV-2 nucleocapsid protein and offer rapid results in 15–30 min, facilitating frequent community-based testing [17,18,19,20]. The currently available SARS-CoV-2 diagnostic tests provide information regarding current infections that can inform appropriate treatments and facilitate the efforts of public health departments to counteract the pandemic. Although real-time reverse transcription PCR (RT-PCR) is the gold standard method for COVID-19 diagnosis, the Omicron pandemic has created a bottleneck due to the surge and increased testing demands. Scaling up COVID-19 testing will require additional resources, supplies, equipment, and alternative protocols to allow nucleic acid testing to continue in the face of potential shortages. We previously reported direct RT-PCR, an alternative to conventional PCR, which is a high-throughput method and achieves comparable sensitivity in a shorter turnaround time [21].

Since 2020, rapid antigen tests (RATs) have been implemented in many countries as diagnostic tools to rapidly screen for SARS-CoV-2 infections on-site during the COVID-19 pandemic. Compared to lab-based nucleic acid amplification tests (NAATs), RATs offer results in 15–30 min without specialized equipment, enabling frequent decentralized testing to quickly identify and isolate infected individuals. Numerous lateral flow and immunochromatographic RATs for detecting the SARS-CoV-2 nucleocapsid protein received emergency use authorization, facilitating widespread community and self-testing [22,23,24]. At-home rapid COVID-19 antigen tests (at-home tests) are a convenient and accessible alternative to determine infection status [24,25]. However, there are some concerns about false-positive results in asymptomatic individuals, which have influenced the clinical use of these rapid tests [26,27]. Given this, the Taiwan Central Epidemic Command Center (CECC) recommend that people who test positive for SARS-CoV-2 infection using an at-home rapid antigen test must undergo a NAAT for confirmation. The at-home kits are provided by basic-level clinics or local public health centers for those suspected to have COVID-19 infection. With the emergence of the SARS-CoV-2 Omicron variants and the increased use of COVID-19 at-home tests, laboratory-based standard processes to confirm the results of these at-home tests are mandatory to inform the decisions of health officials and hospital staff on the treatment or quarantine of individuals.

Taiwan has faced a surge in COVID-19 cases due to the Omicron pandemic since May 2022, with numerous demands for RT-PCR testing leading to a burden on the testing capacities of laboratories. People who test positive using at-home RAT could undergo RT-PCR testing to confirm the diagnosis, significantly reducing the emergency demand. The reliability of at-home RATs as a screening tool in high-incidence areas and the true positive rate remains to be elucidated. Therefore, patients suspected to be positive for SARS-CoV-2 infection through at-home RATs still require confirmation by NAAT. In some instances, the need for confirmatory RT-PCR testing causes an overload of laboratory testing and a delay in timely results. To overcome this situation, implementing optimized COVID-19 testing protocols to increase COVID-19 testing capacity is imperative [28]. In this study, we provide evidence for the use of direct PCR for confirming COVID-19 diagnosis in individuals with a positive at-home rapid antigen test (RAT), and we validated its potential for routine clinical use. Widespread adoption of self-administered RATs enables frequent community screening but requires follow-up confirmatory testing. Direct PCR allows reverse transcription and PCR amplification directly from samples without nucleic acid extraction, increasing throughput.

## 2. Materials and Methods

### 2.1. Study Design and Clinical Specimens

The Tri-Service General Hospital is located in northern Taiwan. The total population of this area (including Taipei City and New Taipei City) is about 6.6 million. The disease prevalence in the COVID-19 Omicron outbreak in May 2022 was around 11–17%, as reported by the Taiwan Center for Disease Control. Taiwan Food and Drug Administration (FDA)-authorized at-home COVID-19 diagnostic RATs were provided for free by local health departments or purchased at nearby pharmacies (https://covid19.mohw.gov.tw/en/cp-5281-63752-206.html, accessed on 1 October 2023). During the study period, eligible patients (*n* = 5652), aged 5 to 99 years, who visited the emergency outpatient department seeking SARS-CoV-2 confirmation or medical treatment between 1 May and 31 May 2022 were recruited. A nasopharyngeal swab was obtained from these eligible subjects using a standard procedure. These specimens were analyzed by NAATs performed on the LabTurbo AIO 48 system (LabTurbo, New Taipei City, Taiwan), an automated high-throughput sample-to-results diagnostic platform. We defined positive cases by RT-PCR targeting *E* and *N1* genes of SARS-CoV-2 as the reference method. Residual viral transport medium (VTM) was analyzed by direct RT-PCR conducted on a LightCycler 96 instrument (Roche, Mannheim, Germany). We focused on the subjects who tested positive with an at-home RAT with a different confirmatory test to compare the clinical performance of the direct PCR method. This study was approved by the Institutional Review Board of the Tri-Service General Hospital (TSGHIRB No.: C202005041). Informed consent was obtained from all study participants.

### 2.2. RT-PCR Testing for SARS-CoV-2 Detection

Samples were tested for SARS-CoV-2 as described previously [29]. Briefly, SARS-CoV-2 RT-PCR testing was performed using the LabTurbo AIO 48 system (LabTurbo, New Taipei City, Taiwan), a multiplex quantitative RT-PCR kit, including a three-target (E, N1, and RNase P), single-reaction, triplex assay used for SARS-CoV-2 detection. This assay maximizes specificity by targeting multiple distinct viral genes. The E gene assay detects the viral envelope, while the N1 assay targets nucleocapsid protein. Human RNase P serves as an internal control. Amplification curves are analyzed by the accompanying QPCR software with threshold cycles <40 for E, N1, and RNase P indicating a positive result. The inclusion of multiple gene targets compensates for mutations in emerging viral variants that could compromise detection by assays targeting only a single region. The LabTurbo system allows automated extraction and pipetting, followed by real-time RT-PCR and result interpretation, increasing sample throughput. Overall, this represents a sensitive and specific workflow for qualitative detection of SARS-CoV-2 infection. Ongoing optimization of extraction-free direct PCR methods could build on this rapid, automated testing capacity.

### 2.3. Direct RT-PCR Assay

The direct RT-qPCR diagnosis method without RNA extraction was implemented during the Omicron pandemic from May 2022. We previously proposed the use of the novel direct RT-PCR [21]. Briefly, for the lab-developed direct RT-qPCR, we used 8 μL of VTM collected from nasopharyngeal swabs of the patients. The RT-qPCR mixture (25 µL) contained the template as input material for the LabTurbo AIO COVID-19 RNA Testing Kit (Acov11240, Taipei, Taiwan) followed the protocol of the commercial kit, which detected the *N1* and *E* genes of SARS-CoV-2 and a human gene (*RP*) as an internal control.

### 2.4. RT-PCR Detection of SARS-CoV-2 VOC

The positive SARS-CoV-2 specimens were screened using six multiplex RT-PCR assays, as described previously [30]. Briefly, a 20 μL reaction mixture (5 μL RNA and 15 μL PCR master mix) was designed to detect nine mutations (ΔHV 69/70, K417T, K417N, L452R, E484K, E484Q, N501Y, P681H, and P681R) against the receptor binding domain (RBD) of the SARS-CoV-2 spike protein. These mutations are characteristic of major variants of concern including Alpha, Beta, Gamma, Delta, and Omicron. The presence of mutation-specific amplicons generates fluorescent signals, allowing discrimination between different SARS-CoV-2 variants. This single-reaction multiplex assay provides rapid genotype screening to determine the variant lineage to which a sample belonged.

### 2.5. Point-of-Care Nucleic Acid Testing for SARS-CoV-2

For specimens in which rapid screening positives were inconsistent with direct RT-PCR, in addition to in-house RT-PCR, we also used two point-of-care nucleic acid amplification testing (NAAT) platforms for SARS-CoV-2 detection: the Liat SARS-CoV-2 and Influenza A/B assay (Roche Molecular Systems, Pleasanton, CA, USA) run on the Liat analyzer, and the Cepheid Xpert Xpress SARS-CoV-2/Flu/RSV cartridge assay (Cepheid, Sunnyvale, CA, USA) run on the GeneXpert system. The Liat and Xpert systems allow automated rapid sample processing and real-time RT-PCR, providing results in under 30 min. Both multiplex assays detect SARS-CoV-2 alongside influenza A, influenza B, and/or respiratory syncytial virus (RSV), facilitating differential diagnosis, in addition to using the conventionally RT-PCR as a reference method, as described previously [31].

### 2.6. Statistical Analyses

Analyses were performed using Excel (Microsoft Corp, Redmond, WA, USA) and GraphPad Prism Version 8.0 (GraphPad, Inc., San Diego, CA, USA). We defined the significance level for all analyses as an alpha (α) of 0.05. *p*-values below 0.05 were considered statistically significant. All quantitative data are presented as mean ± standard deviation or median and interquartile range. The statistical tests employed were selected based on the data characteristics and study aims to compare diagnostic assay performance.

## 3. Results

### 3.1. Confirmation of At-Home RAT Positive Results with RT-PCR Test Results

This study included individuals aged 5–99 years who had positive results on self-administered at-home rapid antigen tests (RATs) for SARS-CoV-2. The estimated prevalence of SARS-CoV-2 infection, as determined by RT-PCR, was approximately 11–15% in Northern Taiwan during the study period, which coincided with the Omicron variant wave of the COVID-19 pandemic. In total, 5652 subjects from the emergency department outpatient clinic for epidemic prevention were included. Of these, 5290 tested positive for SARS-CoV-2 by RT-PCR, confirming the initial RAT results. However, 362 RAT-positive specimens were confirmed as negative for SARS-CoV-2 infection by RT-PCR, indicating false-positive RAT results (Table 1; Figure 1A). The cycle threshold (Ct) values of the positive results by RT-PCR were 16.78 ± 4.30 (mean ± 1STD).

### 3.2. Variant of Concern Real-Time RT-PCR Typing

Of the 5290 specimens that tested positive by RT-PCR, variant genotyping revealed the BA.2.3 sublineage of the Omicron variant (Table 1). These findings demonstrate that the BA.2.3 subvariant was the predominant cause of SARS-CoV-2 infection and had emerged as the dominant variant driving the Omicron wave in Taiwan during the study period. The rapid rise to dominance of BA.2.3 over earlier Omicron sublineages like BA.1 and BA.2 reflects its competitive advantage conferred by mutations enabling higher transmissibility and/or immune evasion. The monitoring of variant prevalence and dynamics in near real-time can inform public health strategies to counter emerging variants of concern and prevent future waves of infection.

### 3.3. Clinical Performance of Direct RT-PCR Assay

The residual nasopharyngeal specimens from RAT-positive subjects were subjected to direct RT-PCR. We analyzed 5652 RAT-positive specimens in this study. The comparison of at-home RAT-positive and direct RT-PCR testing results are shown in Figure 1B. The cycle threshold (Ct) values of the positive results of direct RT-PCR were 17.04 ± 4.19 (mean ± 1STD). The direct RT-PCR showed less sensitivity in samples with lower viral load (CT value > 30) (Figure 1C). After grouping Ct, we compared the two methods for confirming SARS-CoV-2 infection below the Ct value of 30 using a chi-square test (*p* = 0.4462). Therefore, it can be inferred that there is no significant difference between the two methods below the Ct value of 30 (Table 2). The statistical results presented here show that direct PCR is comparable to conventional RT-PCR. The *E* gene-positive specimens (*n* = 5183) confirmed a strong correlation between the results of direct RT-PCR and conventional RT-PCR (Figure 2).

### 3.4. Establishing a Flowchart for Diagnosis of RAT-Positive Patients

The results above showed that direct RT-PCR has comparable sensitivity to conventional RT-PCR, with a shorter turnaround time, of less than 1.5 h, due to the elimination of the RNA extraction process. Therefore, we established a flowchart to obtain a timely and accurate report of SARS-CoV-2 infection status. To decrease the inconsistent results between RAT-positive and direct RT-PCR-negative results due to the lower viral load, we also used point-of-care nucleic acid testing platforms, including Liat SARS-CoV-2 and Influenza A/B and Cepheid Xpert Xpress SARS-CoV-2/Flu/RSV, as a third test to assist the diagnosis (Figure 3). Using this standard diagnostic process, subjects suspected to have COVID-19 infection could obtain timely results and medical treatment.

## 4. Discussion

The COVID-19 pandemic has catalyzed intense biomedical innovation, with emergent diagnostic and therapeutic technologies conferring meaningful public health benefits to mitigate viral spread. While real-time reverse transcription polymerase chain reaction (RT-PCR) represents the current gold standard for COVID-19 diagnosis, reverse transcription loop-mediated isothermal amplification (RT-LAMP) provides an alternative decentralized testing platform with simpler instrumentation needs. Furthermore, the recent surge in confirmed infections has created an urgent need for scaled testing capacities beyond conventional diagnostics. To address this challenge, a myriad of rapid antigen detection kits have been promptly approved for use. These readily available and inexpensive tests offer a valuable first-line screening option, enabling the rapid isolation of positive cases to mitigate transmission.

We evaluated the correlation between outcomes of commercially available rapid antigen tests for home use and conventional reverse transcription polymerase chain reaction (RT-PCR) during the Omicron epidemic in Taiwan. Our data found that from May 1 to 31 May 2022, at-home tests were increasingly used to evaluate COVID-19 status due to the efforts of the Taiwan CECC public health policies to prevent the overburdening of diagnostic laboratory facilities. The use of the at-home tests became more common after the Omicron strain emerged compared to the surge periods of other strains, namely the Delta and Alpha variants. Since the Omicron surge, the COVID-19 incidence has increased considerably to 10–20% in a different region of Taiwan than in the previous B.1.1.7 pandemic [17]. However, a majority of Taiwan has received two complete doses of the COVID-19 vaccine and even a third booster dose [32]. Previous reports have revealed that the Omicron variant demonstrates substantial immune evasion and an ability to reinfect individuals with existing immunity from vaccination or prior infection by other variants of concern (VOC). This immune evasion is conferred by Omicron’s extensive mutations in the spike protein, particularly in the receptor binding domain, which is the target of neutralizing antibodies. Consequently, Omicron infections have been documented even in fully vaccinated individuals and those with previous confirmed Delta variant infections. Real-world evidence indicates significantly reduced vaccine effectiveness against Omicron symptomatic infection compared to prior variants [33,34]. This explains the surge in Omicron cases despite the high vaccine coverage.

While POCT assays hold promise for rapid and accessible SARS-CoV-2 detection, several limitations preclude them from fully replacing direct and conventional RT-PCR assays. These limitations can be categorized into four main areas: performance, throughput, cost, and other relevant metrics. Regarding performance, although POCT may excel in sensitivity in specific scenarios, there is a higher risk of false positives, potentially leading to unnecessary patient isolation and resource allocation. Some POCT assays are designed to detect specific viral variants or mutations, restricting their versatility compared to broader-range assays like direct and conventional RT-PCR. Second, POCT devices are typically designed for individual or small-scale testing, making them impractical for high-volume sample processing, especially in large testing laboratories or during outbreaks. This limitation significantly restricts their utility in resource-constrained settings. Third, the cost per POCT test is generally higher than direct and conventional RT-PCR tests, particularly when factoring in equipment and consumable expenses. This can be a significant burden in resource-limited settings, where cost-effectiveness is crucial. Finally, obtaining regulatory approval for POCT assays can be a complex and time-consuming process, delaying their availability for broader clinical use and potentially limiting their impact on public health initiatives.

The widespread distribution of readily accessible and cost-free at-home testing kits within densely populated regions holds significant promise in mitigating the spread of the severe acute respiratory syndrome coronavirus 2 (SARS-CoV-2). Close monitoring of the efficacy of rapid antigen tests (RATs) in detecting SARS-CoV-2 infection, particularly the highly transmissible Omicron variant, serves as a vital tool for informing the development of effective public health strategies and targeted prevention efforts tailored to specific geographic areas. Moreover, the timely availability of at-home testing empowers individuals to seek prompt diagnosis and access to oral COVID-19 antiviral medications within the crucial five-day window following symptom onset, thereby mitigating the severity of illness and reducing healthcare resource utilization for non-hospitalized patients. By facilitating earlier identification and intervention, widespread utilization of at-home testing can play a pivotal role in curbing the transmission of SARS-CoV-2 and safeguarding public health. It also ensures timely diagnosis and treatment with oral COVID-19 antiviral drugs for non-hospitalized patients within 5 days of symptom onset [35,36]. Moreover, the direct RT-PCR test for SARS-CoV-2 Omicron variants has a shorter turnaround time compared to that of conventional RT-PCR. The direct RT-PCR test is feasible and could be implemented in any diagnostic laboratory without the need for specialized equipment and skilled personnel. Establishing a rapid, accurate, and economic method for SARS-CoV-2 diagnosis is imperative for SARS-CoV-2 infection prevention and control worldwide. Our findings indicate that the direct RT-PCR method can identify the Omicron variant (BA.2), increase the testing capacities of laboratories, and provide accurate testing results.

There are several limitations of this study. First, the commercial kits used for at-home tests were from different manufacturers, which may affect the sensitivity obtained. Second, since only the positive results of at-home tests were confirmed by the RT-PCR method, we do not have a clear overview of false negative results. Third, the samples used in this study were selected on a nonprobability basis; thus, the results might not be generalizable to different age groups. However, our findings particularly demonstrate that the RATs have high accuracy in areas with a high prevalence rate, and could be used as the basis for determining infection with SARS-CoV-2. The timely diagnosis of COVID-19 is fundamental for determining appropriate treatment and patient management strategies.

## 5. Conclusions

This study describes the clinical performance of at-home COVID-19 antigen tests, especially in densely populated areas, and highlights the importance of public health interventions to overcome the Omicron pandemic. The direct RT-PCR method demonstrated greater feasibility and accessibility compared to conventional RT-PCR by eliminating nucleic acid extraction, which increases testing capacity. Our study suggests that at-home rapid antigen tests (RATs) may be no less sensitive than nucleic acid amplification tests (NAATs) in detecting the Omicron BA.2 subvariant. This supports the continued utility of rapid testing for widespread screening, even amid viral evolution. However, false positives were observed with RATs, underscoring the need for confirmatory testing when implementing broad community-based testing programs. Our findings will assist hospitals and health departments in choosing and integrating different diagnostic tools to balance speed, cost, throughput, and sensitivity for controlling the COVID-19 pandemic as new variants emerge. Expanded access to streamlined, high-volume testing, using approaches like direct RT-PCR, will bolster capacity to meet surges in demand. Coupling sensitive confirmatory testing with rapid initial screening will maximize epidemiologic monitoring and the containment of future outbreaks.

## Figures and Tables

**Figure 1 diagnostics-13-03668-f001:**
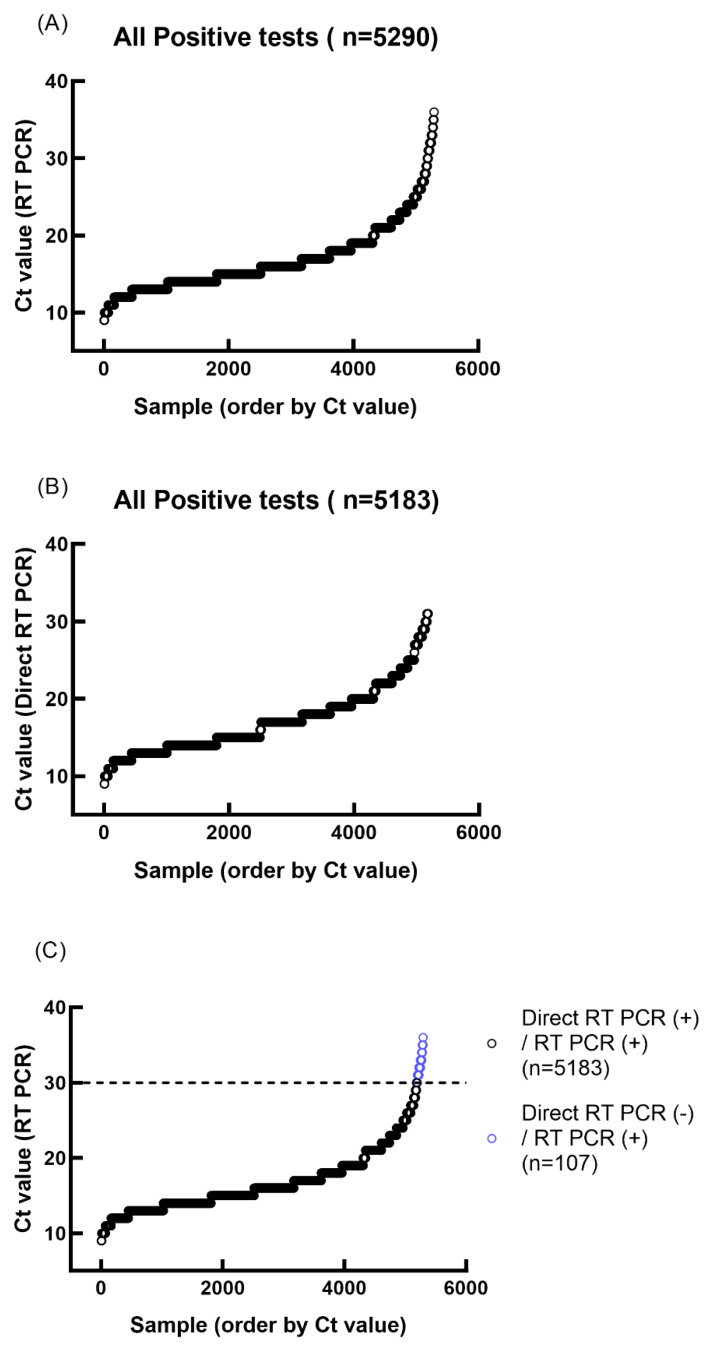
Ct values obtained from RT-PCR and direct RT-PCR of positive at-home rapid antigen test samples. (**A**) RT-PCR Ct value plot ordered by Ct value. (**B**) Direct RT-PCR Ct value plot ordered by Ct value. (**C**) Comparison of RT-PCR and direct RT-PCR test results.

**Figure 2 diagnostics-13-03668-f002:**
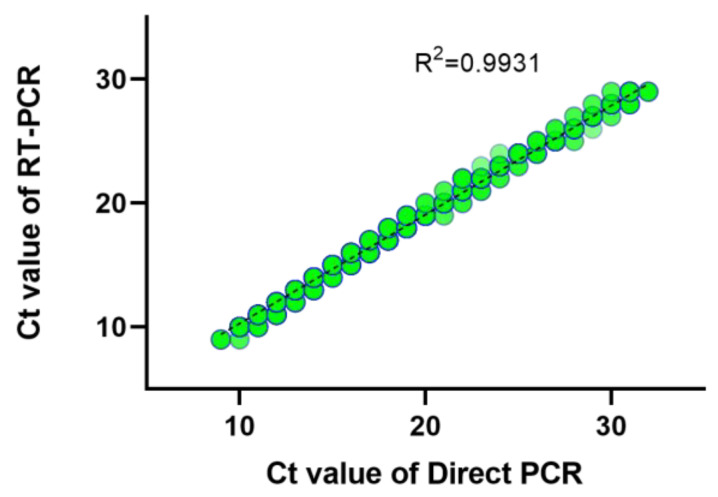
Correlation of Ct values of clinically positive specimens by RT-PCR and direct RT-PCR. The shade of the color represents the number of observations.

**Figure 3 diagnostics-13-03668-f003:**
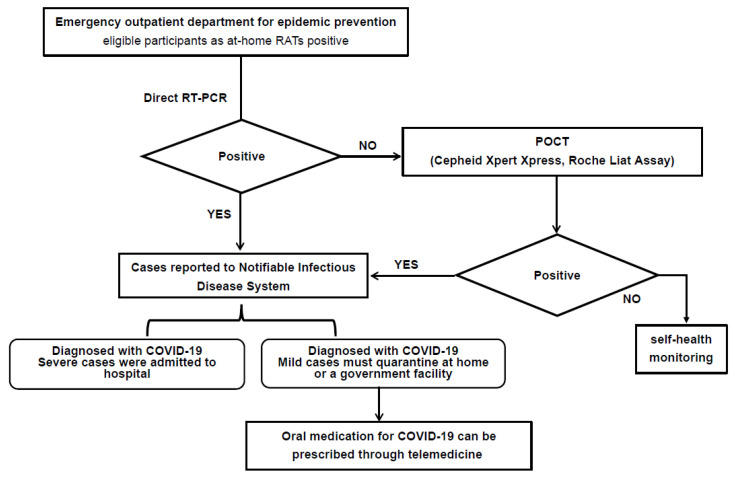
Flow chart for diagnosis of RAT-positive patients. RAT: rapid antigen test; RT-PCR: reverse transcription-polymerase chain reaction; POCT: point-of-care testing.

**Table 1 diagnostics-13-03668-t001:** Demographic characteristics among COVID-19 patients.

	All Patients (*n* = 5652)	
**RT-PCR result**		
Negative, no, (%)	362 (6.4%)	
Positive, no, (%)	5290 (93.6%)	
**All positive patients (*n* = 5290)**		**Ct Value, Mean (SD)**
**Variant**		
Omicron BA.2.3, no, (%)	5290 (100%)	16.78 ± 4.30
**Sex**		
Male, no, (%)	2367 (44.7%)	16.72 ± 4.09
Female, no, (%)	129 (55.3%)	16.84 ± 4.46
**Age, median (SD), y**	66 ± 17.6	
<12, no, (%)	301 (5.7%)	17.64 ± 3.61
12–17, no, (%)	44 (0.8%)	17.75 ± 4.51
18–49, no, (%)	3142 (59.4%)	17.03 ± 4.41
50–70, no, (%)	1544 (29.2%)	16.23 ± 4.11
>70, no, (%)	259 (4.9%)	15.96 ± 4.19

**Table 2 diagnostics-13-03668-t002:** Comparison of direct RT-PCR and conventional RT-PCR.

Ct Value	Conventional RT-PCR (*n* = 5652)	Direct RT-PCR (*n* = 5652)	*p* Value
<11, no, (%)	68 (1.2%)	59 (1.1%)	0.4462
11–20, no, (%)	4277 (75.7%)	4247 (75.1%)	
21–30, no, (%)	853 (15.1%)	858 (15.2%)	
>30, no, (%)	92 (1.6%)	19 (0.3%)	
Negative, no, (%)	362 (6.4%)	469 (8.3%)	

## Data Availability

The data presented in this study are available on request from the corresponding author. The data are not publicly available due to the need to protect patient privacy.

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
