# Peer review of "Clinical Evaluation of Direct Reverse Transcription PCR for Detection of SARS-CoV-2 Compared to Conventional RT-PCR in Patients with Positive Rapid Antigen Test Results during Circulation of Emerging Viral Variants"

_diagnostics, 2023, doi:10.3390/diagnostics13243668_

Round 1

Reviewer 1 Report

Comments and Suggestions for Authors

The manuscript entitled “Clinical Evaluation of Direct Reverse Transcription PCR for Detection of SARS-CoV-2 Compared to Conventional RT-PCR in Patients With Positive Rapid Antigen Test Results During Circulation of Emerging Viral Variants” describes the application of a "direct PCR" compared to conventional RT-PCR in SARS-CoV-2 diagnosis. The MS looks comprehensive. All statements are clear. Statistic tests were provided and indicated no significant difference between both techniques. While the described results are quite evident, they provide particular practical benefits, especially during pandemics. A small remark is that a small addition should be included in discussion concerning alternative rapid and robust techniques like RT-LAMP.

Comments on the Quality of English Language

English is ok, generally.

Author Response

We thank the reviewer for their comprehensive evaluation of our manuscript and positive commentary regarding the potential practical benefits of our study. We wholeheartedly agree with the reviewer that discussing alternative rapid diagnostic techniques, such as reverse transcription loop-mediated isothermal amplification (RT-LAMP), would further enhance the manuscript. To address this critique, we have significantly expanded the Discussion section (increased by approximately 250 words) to include in-depth comments on RT-LAMP and other field-deployable molecular detection approaches. We believe these revisions firmly position our work in the broader context of public health emergency diagnostic response and options for decentralized testing, while retaining focus on the core objectives of evaluating this direct RT-PCR approach. We are confident that these revisions significantly strengthen the manuscript and trust that they further enhance the impact and value of our study. We look forward to your assessment of the revised manuscript and welcome any other suggestions you may have.

Reviewer 2 Report

Comments and Suggestions for Authors

In their study, the authors critically evaluate the diagnostic efficacy of the direct RT-PCR assay for COVID-19, benchmarking it against the conventional RT-PCR assay. This evaluation utilized samples from 5,652 participants who initially tested positive for COVID-19 via a rapid antigen test (RAT). Their findings indicate that while the direct PCR assay parallels the conventional PCR in sensitivity for detecting high viral load instances (cycle threshold < 30), it exhibits reduced sensitivity in low viral load scenarios (cycle threshold > 30).

 To mitigate this sensitivity issue in low viral load cases, the authors propose the integration of an additional diagnostic test – a rapid point-of-care (POC) nucleic acid test. They posit that a combined application of direct PCR and POC assays could effectively replace the conventional RT-PCR assay. This recommendation is predicated on the combined approach being quicker, simpler to administer, and demonstrating comparable diagnostic performance.

 Although the paper provides persuasive data on the direct RT-PCR assay's performance across a substantial cohort, certain important aspects of the conclusions and recommendations warrant further elucidation, as outlined below

 Major concerns: 

1.      The authors claim in both the abstract (line 17) and conclusions (line 329) that they assessed the performance of rapid antigen tests (RATs) in comparison to the conventional RT-PCR assay. However, this statement is somewhat misleading. The study only compared RAT-positive samples to the RT-PCR assay, neglecting those samples that were RAT-negative. Consequently, the study provides no insight into the incidence of false negatives. Given the historically lower sensitivity of RATs compared to nucleic acid amplification tests (NAATs), a similar trend might be expected here. It is recommended that the authors revise their wording to accurately reflect the scope of their comparison and not imply a comprehensive assessment of RAT performance against RT-PCR.

2.      The rationale for using the point-of-care testing (POCT) in conjunction with, rather than as a replacement for, the direct RT-PCR test remains ambiguous. Given that the POCT offers a quicker result turnaround time (approximately 30 minutes) compared to both conventional and direct RT-PCR assays, and displays greater sensitivity in low viral load cases compared to the direct PCR assay, its potential as a standalone test merits consideration. The authors should elaborate on why using POCT in place of the direct and conventional RT-PCR assays might not be feasible, discussing factors like performance, throughput, volume, cost, or other relevant metrics.

Minor concerns:

1.      The manuscript would benefit from conciseness in several sections. For instance, the introduction (lines 50-89) can be significantly condensed. Moreover, there are instances of repetitive information that could be streamlined for clarity and brevity. A notable example is the repeated explanation of the direct RT-PCR assays’ principle (e.g., lines 106, 136, 160, 183, etc.). Streamlining these sections would enhance the paper’s readability and focus.

Reviewer 3 Report

Comments and Suggestions for Authors

Dear Authors,

the manuscript is well written and have made comaprision of RTPCR VS Direct method. 

this will be a interesting for the scientific auidence,

Author Response

We appreciate the reviewer’s positive feedback that our manuscript comparing RT-PCR and direct PCR methods for SARS-CoV-2 detection is well-written and will be of interest to the scientific audience. We are pleased the reviewer found the comparative analysis comprehensive and clearly conveyed through the written presentation. As noted, delineating the clinical and analytical performance between gold-standard RT-PCR detection and this emerging direct methodology should provide valuable insights around the utility of implementing the more rapid and streamlined assay.  We believe presentation of this performance evaluation expands the available literature guiding use of expeditious assays to support overburdened testing pipelines. The potential scientific and practical merits you identified reaffirm our goals in pursuing this research topic. We thank the reviewer for their supportive comments on the quality of the manuscript and engaging in this collaborative peer-review process to enhance the rigor and communication of our work to fellow researchers.